Serum metabolite profiling of a 4-Nitroquinoline-1-oxide-induced experimental oral carcinogenesis model using gas chromatography-mass spectrometry

Ge Shuyun 1
Zhou Haiwen 1
Zhou Zengtong 1
Liu Lin 2 3
Lou Jianing 18939829711@163.com 2 3 4
1 Department of Oral Medicine, Shanghai Key Laboratory of Stomatology, Shanghai Ninth People’s Hospital, Shanghai Jiao Tong University School of Medicine , Shanghai , P.R.China
2 Jiangsu Key Laboratory of Oral Diseases, Nanjing Medical University , Nanjing , P. R. China
3 Department of Oral Medicine, Affiliated Hospital of Stomatology, Nanjing Medical University , Nanjing , P. R. China
4 Department of Stomatology, Shanghai General Hospital of Shanghai Jiao Tong University , Shanghai , P. R. China
Sharma Gaurav
Electronic publication date: 2021 Jan 4
Publication date: 2021
Volume: 9
Electronic Location ID: e10619
Received 2020 Jun 24; Accepted 2020 Nov 30
Copyright: ©2021 Ge et al.
Copyright year: 2021
Copyright holder: Ge et al.
License: This is an open access article distributed under the terms of the Creative Commons Attribution License, which permits unrestricted use, distribution, reproduction and adaptation in any medium and for any purpose provided that it is properly attributed. For attribution, the original author(s), title, publication source (PeerJ) and either DOI or URL of the article must be cited.
License URL: https://creativecommons.org/licenses/by/4.0/

Keywords: GC-MS, Oral carcinogenesis, Metabonomics, Metabolites, Tricarboxylic acid cycle

Funding: Youth Project of the National Natural Science Foundation of China 30700944 Shanghai Municipal Health Bureau 2012092, ZHYY-ZXYJHZX-201612 This study was supported by the Youth Project of the National Natural Science Foundation of China (No. 30700944), and a grant from the Shanghai Municipal Health Bureau (No. 2012092, ZHYY-ZXYJHZX-201612). The funders had no role in study design, data collection and analysis, decision to publish, or preparation of the manuscript.

==============================
Background

Oral cancer progresses from hyperplastic epithelial lesions through dysplasia to invasive carcinoma. The critical needs in oral cancer treatment are expanding our knowledge of malignant tumour progression and the development of useful approaches to prevent dysplastic lesions. This study was designed to gain insights into the underlying metabolic transformations that occur during the process of oral carcinogenesis.

Methods

We used gas chromatography-mass spectrometry (GC-MS) in conjunction with multivariate statistical techniques to observe alterations in serum metabolites in a 4-Nitroquinoline 1-oxide (4NQO)-induced rat tongue carcinogenesis model. Thirty-eight male rats were randomly divided into two groups, including the 4NQO-induced model group of 30 rats and the healthy control group of five rats. Animals were sacrificed at weeks 9, 13, 20, 24, and 32, post-4NQO treatment. Tissue samples were collected for histopathological examinations and blood samples were collected for metabolomic analysis. Partial least squares discriminate analysis (PLS-DA) models generated from GC-MS metabolic profile data showed robust discrimination from rats with oral premalignant and malignant lesions induced by 4NQO, and normal controls.

Results

The results found 16 metabolites associated with 4NQO-induced rat tongue carcinogenesis. Dysregulated arachidonic acid, fatty acid, and glycine metabolism, as well as disturbed tricarboxylic acid (TCA) cycle and mitochondrial respiratory chains were observed in the animal model. The PLS-DA models of metabolomic results demonstrated good separations between the 4NQO-induced model group and the normal control group.

Conclusion

We found several metabolites modulated by 4NQO and provide a good reference for further study of early diagnosis in oral cancer.

Introduction

Oral cancer is a type of malignant tumor with high degree of malignancy. Oral cancer accounts for 7% of all new cancer cases worldwide, and 270,000 cases annually (Rivera, 2015). Over the past several years, the incidence of oral cancer has increased, especially in developing countries such as Sri Lanka, India, Pakistan and Bangladesh, where it accounts for up to 25% of all cancers (Krishna et al., 2013). Despite the development of surgery, radiotherapy, and chemotherapy in recent years, the five-year survival rate of oral cancer is about 50%. Moreover, post-operative patients experience morphologic changes and different degrees of loss of function. Patients may experience significant dysfunction in talking, swallowing, with alteration of cosmetic appearance, and sensory impairment, as well as chronic pain. Additionally, some healed patients also relapse with a few years of treatment. All these factors when compounded lead to poor mental health.

Oral leukoplakia is acknowledged as a common precancerous lesion. The canceration rate of oral leukoplakia is 0.13%–34% (Warnakulasuriya & Ariyawardana, 2016). Oral leukoplakia is an important source of oral squamous cell carcinoma (OSCC). The World Health Organization (WHO) has considered the control and prevention of oral premalignant lesions and oral cancer as one of the top urgent tasks in the”21st century global oral health strategy”. Thus, screening biomarkers of malignant transformation of oral leukoplakia is important for preventing oral cancer.

Metabolomics is a discipline focused on the changes to endogenous metabolites of biological systems following perturbation by a stimulus (such as a specific gene mutation or environmental change). The formation and development of tumors may be due to changes in the metabolic pathways of cells, which results in the loss of normal cell growth regulation and abnormal proliferation. Metabolomics uses high-throughput, high sensitivity and high precision instrumentation to dynamically analyze the changes in metabolites and identify cells as “cancer - precancerous - normal”. Thus, Metabolomics detects the metabolic markers indicative of the formation and development of cancer. The concentration of proteins, peptides, and metabolites regulating cell growth in multiple cancers were differentiated in tissue fluid than cells (Armitage & Ciborowski, 2017; Kaushik & DeBerardinis, 2018). These minute changes in gene and protein expression are amplified at the metabolite level. Thus, in contrast to gene chip and proteomic approaches, metabolomics is more likely to detect biomarkers associated with cancer(Olivares et al., 2015; Armitage & Ciborowski, 2017; Beger, 2013).

Using a high-resolution testing instrument (such as HR-MAS NMR, High-Resolution GC-MS, High-Resolution LC-MS, et al.) and multivariate statistical analysis methods, metabolomics can accurately identify metabolites that change over the course of cancer development and can be used to evaluate tumors (Blekherman et al., 2011; Bathen et al., 2010; Wen et al., 2010). 1HNMR combined with partial least squares-discriminant analysis (PLS-DA) were used to identify differential metabolites among different groups which is refers to the different developmental stages of chemically induced oral cancer. This approach was used to determine that the levels of valine, lactic acid salt, alanine, and citric acid salt were reduced in oral cancer cells compared to healthy controls, whereas signals arising from glucose, pyruvate, acetone, acetoacetate, 3-hydroxybutyrate and 2-hydroxybutyrate (Wu et al., 2010; Markuszewski et al., 2010; Kim et al., 2010). GC-MS has high sensitivity and high specificity, However, less applied to studies of oral cancer.

The 4-Nitroquinoline 1-oxide (4NQO)-induced carcinogenesis model is widely used for studies of oral cancer development and for screening cancer chemopreventive drugs. In this study, we utilized a gas chromatography-mass spectrometry (GC-MS)-based serum metabolite profiling approach to discriminate 4NQO-induced oral carcinogenesis from normal controls at different times, and to monitor the changes in metabolites between such samples.

Materials and Methods

4NQO-induced rat carcinogenesis and sampling

Rats were obtained from Shanghai Sippr-BK laboratory animal Co. Ltd. A total of 38 Wistar rats with origins in closed groups were used at 160 days old, and 220 ± 10 g in body weight. Rats were fed full-value nutritional granulated fodder and raised in separate cages (four rats per cage). Rats were transferred to the holding room under controlled conditions, with a temperature of 23 ± 2 °C, and 30–50% humidity. 4NQO (Sigma, USA) was dissolved in distilled water to a concentration of 0.002% and was placed in a brown bottle at 4 °C until use. All animals and experimental procedures were approved by the Management Committee of Animal Experimental Ethical Shanghai Ninth People’s Hospital affiliated to the Shanghai Jiao Tong University, School of Medicine HKDL (2016) 2.

The 38 rats were divided into two groups as follow. A: control group, given water (n = 5); B: model group, given 4NQO (n = 33).

The criteria were established for euthanizing animals prior to the planned end of the experiment. Rats were sacrificed at 9, 13, 20, 24, and 32 weeks from the beginning of the experiment to collect the tissues Representing temporal carcinogenesis progression that demonstrates multiple dysplastic, preneoplastic, and neoplastic lesions after long-term treatment. The mice were sacrificed by intraperitoneal injection of sodium phenobarbital. Animal carcasses were being loaded into the garbage bag and handled by Institute of Laboratory Animals. We collected serum samples and stored it at −80 °C for GC −TOF mass spectrometry. Besides, we collected the tissue samples for histopathological examination.

Pathological examinations

Gross lesions were identified and photographed. The histological determinations of squamous neoplasia were performed by a pathologist on the sectioned tissue samples. The sections from the tongues were deparaffinized, rehydrated, and stained with hematoxylin-eosin (HE) for histopathological analyses. The lesions were classified into four types: mild epithelial dysplasia, moderate epithelial dysplasia, severe epithelial dysplasia, and squamous cell carcinoma (SCC). Epithelial lesions were diagnosed according to the criteria described by the WHO (Leininger & Jokinen, 1994; Kramer et al., 1978).

GC-MS spectra acquisition of serum samples and data pretreatment

As the procedure we previously described, serum metabolites were analyzed by chemical derivatization and slightly modified (Chen et al., 2011; Wei et al., 2012). We added 100 µL aliquot of serum to two internal standard solutions, 10 µL L-2-chlorophenylalanine in water (0.3 mg/mL) and 10 µL heptadecanoic acid in methanol (1 mg/ml). Then, we swirled it for 10 s. We extracted a mixture with 300 µL of methanol/chloroform (3:1) in eddy 30 s. We stored the samples at −20 °C for 10 min and centrifuged it at 12,000 g for 10 min. Then, we transferred 300 µL supernatant to an aliquot for vacuum drying at room temperature. The residue was derived using a two-step method. First, we added 80 µL methoxyamine (15 mg/mL in pyridine) to the bottle and incubated it at 30 °C for 90 min. Then 80 µL BSTFA (1% TMCS) was added and incubated at 70° C for 60 min.

We injected each 1 µL derived solution into antae 6890N gas chromatography in a non-splitting manner in combination with Leco Pegasus IV time-of-flight MS instrument (Leco, St. Joseph/MI, USA). Run alternately in control-model-control order to reduce system analysis bias. Db-5ms capillary column (30 m × 250 µm i.d., 0.25 µm film thickness; (5%-phenyl)-methylpolysiloxane bonded and cross-linked; Agilent J&W Scientific, Folsom, CA) was used for separation. Helium was carrier gas with a constant flow rate of 1.0 mL/min. The injection temperature was set at 270 °C, the transfer interface temperature at 260 °C, and the ion source temperature at 200 °C. We programmed GC temperature as 80 °C isothermal heating for 2 min Then the oven temperature was rise to 180 °C by 10 °C /min, 240 °C by 5 °C /min, and 290 °C by 25 °C /min. Finally, it was kept at 290 °C for 9 min. Electron shock ionization (70 eV) in a full scan mode (M/Z30-600) was used to obtain 20 spectra /s in an MS setting.

GC-MS data analysis

The data analyzed by GC-MS was converted to NetCDF format through the data analysis interface of PE instrument (PerkinElmer Inc., Waltham, MA, USA). Subsequently, we extracted each file by a custom script in MATLAB 7.0 (The MathWorks, Inc., Natick, MA, USA) to perform a data preprocessor that included baseline correction, peak deconvolution and calibration, exclusion of internal standard (I.S.) and solvent peaks, and normalization to the total ion current. The output data were organized in the form of any arbitrary peak index (retention time-m/z pairs), sample names (observations), and peak intensity information (variables). Internal criteria and any known artificial peaks, such as those caused by noise, column bleeding, and ethyl chloroformate derivative processes, were removed from the data set. Partial least-squares discriminant analyses (PLS-DA) were performed by SIMCA-P 12.0 software (Umetrics, Umeå, Sweden). Based on a variable importance in projection (VIP) threshold (VIP > 1) from the 7-fold cross-validated PLS-DA model, a number of metabolites responsible for the differentiation of metabolic profiles of the 4NQO-induced model group and the normal control group could be obtained. Meanwhile, through the parametric student t test and non-parametric Wilcox-Mann–Whitney test in the Matlab statistical toolbox, we verified the metabolites identified by the PLS-DA model at the univariate level, and the critical value was set at 0.05. The corresponding fold changes showed the differences of these selectable metabolites between the 4NQO-induced model group and the normal control group. Additionally, compound identification was performed by comparing the mass fragments using the NIST 05 Standard mass spectral databases in NIST MS search 2.0 (NIST, Gaithersburg, MD) software with a similarity of more than 70%, verified using available citric acid reference compounds.

Results

Histological changes of the mucosa following 4NQO administration

Rats showed no overt ill effects from the carcinogen applications. The body masses of control and experimental animals remained similar until the end of the experiment, when the 4NQO-treated animals tended to be lighter than the controls. Such findings were likely due to the decreased food intake in rats with oral carcinomas.

The initial change of mild epithelial dysplasia was observed at 9 weeks following 4NQO treatment. The thickness of the spinous cell layer gradually increased and occasionally exhibited disordered basal cells. After 13 weeks post-4NQO treatment, further dysplastic changes occurred, from moderate epithelial dysplasia, to severe epithelial dysplasia, and ultimately, invasive squamous cell carcinoma (SCC) being observed in the majority of animals (Figs. 1A–1E). Histological changes in each group were shown in Table 1. Typical histopathological results of oral lesions caused by 4NQO were shown in Fig. 1.

Figure 1 Histopathological images from 4NQO-induced oral lesions (×200).

(A) Normal squamous epithelium of the tongue. (B) Mild epithelial dysplasia. (C) Moderate epithelial dysplasia. (D) Severe epithelial dysplasia.(E) Squamous cell carcinoma (SCC).

Table 1 Histological changes in rat tongue by 4NQO treatment.

Group
(week)	No. of normal	Mild epithelial dysplasia	Moderate epithelial dysplasia	Severe
epithelial
dysplasia	No. of SCC	Incidence
(%)	
control	5	0	0	0	0	0	
9week	0	4	3	0	0	0	
13week	0	1	5	1	0	0	
20week	0	0	5	0	0	0	
24week	0	0	0	2	4	33.3	
32week	0	0	0	0	8	100	

Metabolic profiles between the model group and controls

First, we set the Y value of normal group samples to 0, while that of the model group was set to 1, join Y2 (sampling time point) and Y3 (pathological results). With the three guidance variables, we established a PLS-DA model for multivariate statistical analysis. The separation trends and sampling time points were closely linked, and thus, we observed the physiological changes of the normal group. The result of the PLS-DA model revealed the separation of rats with SCC and dysplasia from the normal group. The PLS-DA model demonstrated satisfactory modeling and predictive abilities using one predictive component and two orthogonal components (R2X = 0.397, R2Y = 0.982, Q2 = 0.762), achieving a distinct separation between the metabolite profiles of the 4NQO-induced model group and normal control group (Fig. 2).

Figure 2 Score plot for the PLS-DA model of the GC-MS spectral data from the model group and the healthy control group.

We used MS spectrum database to identify 16 metabolites and confirmed the reference standard between differential variables by using VIP values (VIP >1) in the PLS-DA model and the Wilcoxon-Mann–Whitney test (p < 0.05) (Table 2). Among the identified metabolites in the model group, glycolysis showed the most significant change compared to controls. The altered serum metabolites included elevated lactic acid, citric acid, pyruvate, arachidonic acid, glycine, and decreased 3′, 5′-cyclic dGMP, PGF2alpha dimethyl amide, and 1-hexadecyl-2-acetyl-glycero-3-phosphocholine in the model group, compared to normal controls.

Table 2 List of Identified Differential Metabolites between Model Group and Healthy Controls.

No	Metabolitea	FCb (model/control)	VIPc	pd	
1	PGF2alpha dimethyl amide	−1.37	1.63	0.0138	
2	3′, 5′-Cyclic dGMP	−1.03	1.42	0.042	
3	lactic acid	2.84	1.55	0.0036	
4	Palmitoylcarnitine	1.04	1.53	0.0243	
5	Demissidine	1.48	1.55	0.0383	
6	16a-Hydroxydydrogesterone	1.61	1.54	0.0069	
7	arachidonic acid	2.21	1.91	0.0028	
8	Deoxyuridine monophosphate (dUMP)	1.15	1.79	0.0159	
9	oleamide	1.78	1.7	0.0078	
10	citric acid	1.85	2.14	0.0174	
11	1-hexadecyl-2-acetyl-glycero-3-phosphocholine	−1.25	2.05	0.0162	
12	glycerol	1.89	1.65	0.022	
13	glycine	1.497	1.64	0.0056	
14	hexadecanoic acid	1.518	1.52	0.0101	
15	pyruvate	1.832	1.66	0.00123	
16	eicosatetraenoic acid	1.18	1.84	0.0239	
Notes.

a Metabolites verified by reference compounds, other were directly obtained from library searching.

b Fold change was calculated from the arithmetic mean values of each group. Fold change with a positive value indicates a relatively higher concentration present in model group while a negative value means a relatively lower concentration as compared to the healthy controls.

c Variable importance in the projection (VIP) was obtained from PLS-DA with a threshold of 1.0.

d p value was calculated from student t test.

Discussion

4-Nitroquinoline 1-oxide-induced experimental oral carcinogenesis

Several animal models for oral SCC were previously generated, including those for hamsters, rats, and mice.The 4NQO-induced tongue carcinogenesis model is anatomically more similar to human SCC, more convenient, and likely more reproducible, as the carcinogen is administered in the drinking water. The use of 4NQO resulted in a transient model of cancer progression that showed a variety of dysplasia, precancerous and neoplastic lesions after long-term treatment. These continuously changing epithelial cells mimic the transformation of human oral tumors. In addition, 4NQO-induced oral cancer in mice is similar to human head and neck squamous cell carcinoma, affecting the expression of many genes involved in human tumorigenesis. Thus, this animal model is widely used for studies of oral cancer development and to screen chemopreventive drugs(Kanojia & Vaidya, 2006).

4-Nitroquinoline 1-oxide is a water-soluble quinoline derivative that can cause DNA adduct formation, resulting in adenosine substitution for guanosine. 4NQO also undergoes redox cycling to produce reactive oxygen species that result in mutations and DNA strand breaks.

Metabolite variations in the 4NQO-induced oral carcinogenesis model

Metabolomics has great potential for the early detection and mechanisms of oral cancer. In our study, the serum metabolome results of 4NQO-induced tongue carcinogenesis at weeks 9, 13, 20, 24, and 32 showed progressive metabolic disturbance in the serum metabolome during 4NQO-induced oral carcinogenesis. Our study, based on the metabolomics of GC-MS, showed 16 differentially expressed serum metabolites in the model group. The derived PLS-DA model showed a good separation between the model group and the controls.

Arachidonic acid (ARA) is an n-6 essential fatty acid and also a major constituent of biomembranes. ARA is released from membranes by phospholipase A2 and converted into various lipid mediators that exert many physiological actions. ARA can be converted (1) by cyclooxygenases (COXs) to prostaglandin E2 (PGE2) and thromboxane (TxA2); (2) by mammalian lipoxygenases (LOXs) to four major hydroxyeicosatetraenoic acid (5-, 8-, 12-, and 15-HETE); and (3) by cytochrome P450 to epoxyeicosatrienoic acids (EETs) and HETEs. Many studies have shown that lipid mediators derived from ARA, such as prostaglandin E2 (PGE2) and leukotrienes, and the enzymes involved in their production, such as cyclooxygenases (COXs), lipoxygenases (LOXs), and cytochrome P450, are centrally involved in apoptosis and angiogenesis.

The metabolism of ARA plays a role in the pathogenesis of many human cancers (Sakai et al., 2012; Ninomiya et al., 2013; Yarla et al., 2016). Abnormally high levels of serum ARA during the process of oral carcinogenesis suggests that a metabolic imbalance causes the high levels of proliferative eicosanoids, such as PGE2, that are commonly found in tumor cells.

Lactic acid is the final product of the glycolysis pathway, which participates in energy metabolism. The level of lactic acid and fructose in rats with SCC and dysplasia were elevated, compared to the normal group. Lactic acid accumulation may be indicative of enhanced glycolysis. Enhanced glycolysis is considered a common feature of tumors, which is called the ”Warburg effect” (Warburg, 1956). Lactic acid levels were significantly elevated in head and neck cancer, among others (Dart, 2016; Ohashi et al., 2017). In some cancers, pyruvate can be converted into lactic acid under aerobic conditions (Choi et al., 2013). Fructose can also participate in the glycolysis pathway through 6-phosphate glucose and 1, 6-fructose diphosphate. High intakes of fructose were related to elevated risks of colorectal and pancreatic cancers (Goncalves et al., 2019; Aune et al., 2012). The elevation of lactic acid and fructose in rats with SCC and dysplasia suggested that those rats must consume more energy than normal rats.

Pyruvate and citric acid were higher in the serum of squamous cell carcinoma and dysplasia rats. This possibly due to the need for more glycolysis to meet the increased energy requirements of the cell, which was associated with accelerated anabolic metabolism. Pyruvate can be converted to acetyl-CoA in the tricarboxylic acid (TCA) cycle, upon which it is transaminated to alanine, or becomes lactate, particularly under hypoxic conditions (Raimundo, Baysal & Shadel, 2011). Citric acid, an originate product in the TCA cycle, was elevated in rats with SCC and dysplasia. Citric acid in human lung cancer was evaluated and compared with healthy controls (Faubert et al., 2017; Hensley et al., 2016). The formation of citrate from acetyl-CoA and oxaloacetate permits a new round of TCA cycling, generating high energy electrons, CO2, and carbon skeletons that can be used for biosynthesis or anaplerosis. Citrate itself can be extruded into the cytosol and converted to acetyl-CoA by ATP citrate lyase (ACLY) for fatty acid synthesis and the generation of biomembranes (Benjamin, Cravatt & Nomura, 2012).

Extracellular high concentrations of pyruvate and citric acid support anabolism through the fumarate-malate-citric acid pathway, resulting in the production of fatty acids and other biochemical precursors in the carcinogenic process. The addition of pyruvate and citric acid can also be used as chest substrates for the TCA cycle to increase energy production.

The levels of glycerol, hexadecanoic acid, palmitoylcarnitine, demissidine and 1-hexadecyl-2-acetyl-glycero-3-phosphocholine in rats with SCC and dysplasia were significantly different than those of the normal group. Those data implicate deregulated lipid biosynthesis in cancer development. Dysregulated lipid metabolism and heightened de novo lipogenesis are established hallmarks of cancer (Glaysher, 2013; Cheng et al., 2018). Tumor cells are characterized by elevated fatty acid synthesis. Tumor cells synthesize fatty acids for the purposes of membrane synthesis and for the generation of lipid signaling molecules to fuel cell proliferation and cancer malignancy.

Significantly higher levels of glycine were observed in the model group compared to controls. Glycine is the precursor for the biosynthesis of proteins, purine, and glutathione. In addition, glycine is converted into sarcosine (N-methylglycine) by glycine N-methyltransferase, and conversely, sarcosine can be converted into glycine by sarcosine dehydrogenase (Locasale, 2013; Li et al., 2013). In recent years, tumor metabolomics studies have shown that glycine metabolism is related to tumor cell proliferation. In the investigation of NCI-60 cell line plate, the uptake and release rates of more than 200 metabolites were measured. Startlingly, lactic acid production and glucose uptake (i.e., the Warburg effect) had nothing to do with cell proliferation. After correlating individual metabolic fluences with cell proliferation, we found that glycine uptake was most closely related to cancer cell proliferation. Isotopic tracers indicated that the cleavage of glycine in the medium was involved in its catabolism, a pathway that has also been shown to be necessary for rapid cell division (Jain et al., 2012).

Knobloch et al. (2019) also used 4NQO- rat oral cancer model to identify 57 differentially or uniquely expressed metabolites. These metabolites were mainly related to Glycolysis and AMPK pathways. Along with targeting metabolic pathways, a comprehensive analysis of their multiple sets of data further confirmed the array of molecular pathways regulated by BRB phytochemicals. However, we did not use an independent method to verify our results. In the process of chemoprevention of oral cancer, the characteristics of metabolites need to be further identified and verified, which will provide more insights for the development of chemical strategies in cancer prevention and treatment.

Conclusion

In summary, we discovered a significant metabolic transformation in an 4NQO-induced oral carcinogenesis model. The metabolic shift was characterized by an increase in glycolysis and a deficiency in the TCA cycle. Other metabolic processes, including nucleic acid and lipid biosynthesis, were also enhanced as part of cancer-associated metabolic reprogramming. Our GC-MS metabolic analysis yielded a PLS-DA model, which showed a good separation between the model group and the control groups.

Supplemental Information

Supplemental Information 1 Datasets

Click here for additional data file.

Additional Information and Declarations

Competing Interests

Author Contributions

Animal Ethics

Data Availability

The authors declare there are no competing interests.

Shuyun Ge performed the experiments, analyzed the data, prepared figures and/or tables, authored or reviewed drafts of the paper, and approved the final draft.

Haiwen Zhou performed the experiments, analyzed the data, prepared figures and/or tables, and approved the final draft.

Zengtong Zhou performed the experiments, analyzed the data, authored or reviewed drafts of the paper, and approved the final draft.

Lin Liu performed the experiments, prepared figures and/or tables, and approved the final draft.

Jianing Lou conceived and designed the experiments, performed the experiments, authored or reviewed drafts of the paper, and approved the final draft.

The following information was supplied relating to ethical approvals (i.e., approving body and any reference numbers):

All animals and experimental procedures were approved by the Animal Experimental Ethical Inspection Shanghai Ninth People’s Hospital affiliated to Shanghai Jiao Tong University, School of Medicine [HKDL(2016)2].

The following information was supplied regarding data availability:

The raw measurements are available as a Supplementary File.

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
