# Peer review of "Serum metabolite profiling of a 4-Nitroquinoline-1-oxide-induced experimental oral carcinogenesis model using gas chromatography-mass spectrometry"

_PeerJ, doi:10.7717/peerj.10619_

## Round 0.1 · original submission · Major Revisions

As you'll learn from the reports, the referees have found several significant methodological and data analysis concerns. Major issues include inadequate statistical analysis and insufficient methodological details. For editorial consideration, the manuscript should be further improved and provided with a detailed point-by-point response to the reviewer's comments with resubmission.

Reviewer 1 ·

Basic reporting

The current manuscript describes the metabolomics based 4-Nitroquinoline-1-
oxide-induced oral cancer. The subject is interesting, especially due to the animal model used, which resembles the human oral cancer than the DMBA induced one. The work undertaken is substantial whereas the manuscript is well organized and written in a concise manner. experimentatal

Experimental design

The experimental design concerning the animals is rather adequate. Although there is a sufficient amount of experimental animals they were assigned to rather limited population groups (n=5 is marginal given also the fact that the experimental animal setup is not very difficult). The chemical analysis setup is well designed and executed but in my opinion the statistical treatment is not proper. Some specific comments can be found below
line 142 please refer the instrumental parameters used, the calibration procedure of the instrument as well as the mass accuracy obtained.
line 147 please refer the parameters used for data pretreatment e.g what was the mass accuracy in order to consider a metabolite as the same between samples or the time shift between peaks that should be aligned etc
line 155 no data appear on the statistical analysis e.g. are the the UV or Pareto treated how did the authors choose the number of the latent variables etc
line 172 did the authors use some kind of weighting the metabolite levels according to the body weight?
line 191 in order to validate the model as well as the metabolites up/down regulated further steps should be taken such as permutation testing ROC curves and FDR base tests. Did the authors use some QC samples in order to check the validity of the chemical and statistical analysis? Did they use some margin for the appearance of the metabolites in the samples e.g. if the metabolite is present to >50% (or any other percentage the authors consider arguably appropriate) of the samples?
line 194 please refer how did the authors identify the metabolites e.g. based on accurate mass or employing also the fragmentation patterns from the EI ionization.
line 264 In my opinion these results should be considered along with the diminished food intake of the rats that developed cancer. This could probably differentiate the rats in a more pronounced way.

Validity of the findings

no comment

Additional comments

line 57 function: please specify what kind of function is lost
line 57 healing:maybe this is healed?
line 73 higher please change the word to differentiated
line 77 high resolution not all NMR or MS instruments can be considered as high resolution and definitely not the HPLC. Please rephrase
line 78 metabonomics: please use only one term. It seems that currently metabolomics is the way to go.
line 107 please change to phenobarbital
line 115 please specify what HE is.
line 139 please correct 270oC, 260oC also leave a space before units. It is recommended to use a non breaking space
line 154 PLSDA is a multivariate analysis technique
line 232 please use only one term in order to avoid confusion (metabolomics / metabonomics)

Reviewer 2 ·

Basic reporting

In this study, PLS DA were used to analyze the GC MS profile of serum using a 4NQO induced oral carcinogenesis rat model and to identify metabolic biomarkers in the development of oral cancer. The data indicates that GC based metabolomic analyses of plasma distinguish between oral premalignant and malignant lesions.

Comments
This is a nicely designed study and the statistical models are appropriate, and the methodology is reliable. The results of the study are not explained well though. The manuscript could be made more interesting by these suggestions.
In the introduction line 75, the authors should mention the difference between metabolomics and metabonomics in a better way to educate the readers.
Line 72-73 needs a reference for the claim.
Line 80-81 states that 1HNMR combined with PLS-DA can distinguish
oral cancer from healthy serum samples. This sentence is ambiguous and needs to be rewritten.
Line 85-86. How is GC MS better than NMR or LC MS or other methods? The claims about sensitivity and specificity needs references.

Experimental design

Spell check Line 139

What is the reason for sacrificing the animals at 9, 13, 20, 24, and 32 weeks from the beginning of the experiment? Explain in the methodology.

How were the metabolites identified? To elaborate on my question, was a metabolite classified as “identified” if it was found to be present in at least two samples of the same group? Include the criteria in the methodology.

Mention about the reference standards used to identify the 16 metabolites in the methods

Validity of the findings

Line 187-188 states that the growth of animal cycles on metabolite changes was greatly influenced, so we ultimately selected fourth and fifth sampling time points samples. I do not understand what the authors mean by this. Explain or remove the sentence.
Line 192; is Q2 really 762?
Line 197; what is the fold change? Specify.
Abstract mentions glycocine. Is it glycine?
The major drawback of this study is that there is no independent method to validate these findings. The authors should compare their results to these similar studies in the discussion.
https://doi.org/10.3892/ol.2014.2619
DOI: 10.3390/metabo9070140

The concentration variations of the discriminant metabolites at various stages of oral carcinogenesis compared to controls should be graphed.

Additional comments

The supplemental file with the excel data is not easily understandable. Descriptions should be given for each data set. Alternatively, the raw data can be uploaded to metaboLights platform if the authors wish to do so. It is fine if they do not want to.

Reviewer 3 ·

Basic reporting

The work of Ge et al. dedicated to the metabolic profiling of serum in animals treated with known compound that causes oral cancer. The authors divided treated animals to the subgroups in respect to the cancer progression, and compared their metabolic profiles with the healthy control. GC-MS methodology was used to perform metabolic profiling, and the data were analyzed using multi- and univariate analyses, that suggested 16 potential biomarkers for the oral cancer diagnostics.
The scientific premise of the work is scientifically sound. The work is of interest, and mostly written in a clear way. However, I have several concerns regarding the experiments design, and statistical methods, as well as main conclusions from this study.

Experimental design

The experimental design is not fully clear for me. The authors said that 38 animals were included to the study, but all results are based on the analysis of 30 animals. What happened to another 8? Did they die, or were excluded due to other reasons? I also did not really understand how the serum collection was performed. Metabolite turnover is a very fast process, so how fast serum samples were collected and frozen after animal death? I am also concerned about usage of sodium phenobarbital for metabolomics studies. Can the authors provide references in favor of this method?
For the statistical analysis the authors used the Student’s t-test and Wilcoxon-Mann-Whitney for differentiated analysis, however, it is not clear what 2 groups were compared, as before this the authors segregated “diseased” group to 5 sub-groups. Based on their PLS-DA results, I actually would recommend them to perform Dunnett’s test, as there are evidences (based on PLS-DA) that the differences between control group and most severe “scc” group can be stronger. It is also will make more sense considering the division to sub-groups. The results of fold-change between each of 5 groups vs control group can be nicely presented in a heat-map view.

Validity of the findings

If the results section is not detailed enough, the discussion section is very long and vague. I recommend the authors to largely reduce the 1st paragraph of the discussion as it has weak connection to the main goal of their study. Next, arachidonic acid as a fatty acid is not usual metabolite for the method of extraction, derivatization and column that authors used. The methodological set-up of the authors is optimized for polar compounds. Indeed, it is possible to detect FAs under these conditions, but I am concerned about repeatability and preciseness of this. Was arachidonic acid confirmed by analytical standard?
My last major comment is about the PLS-DA role in biomarkers discovery written in both conclusion section and abstract. This is the very well-known fact. It does not mean that we do not have to use it, but I think it is weak to claim as the main conclusion. I think, the biological findings of the authors about alterations in citrate, lactate, pyruvate are much more important and exiting. So it is not “PLS-DA could be used to screen potential biomarkers”, it is “metabolic profiling could be…”

Additional comments

Minor points:
1) Typos (no spaces before references, row 63 – no space before quotation, row 86 – dot, row 139 – degree signs)
2) I am not sure that the statement about novelty of the GC-MS application to oral cancer studies is correct. For example, Yang et al. (https://www.ncbi.nlm.nih.gov/pmc/articles/PMC7174902/) used it. It is true that their experimental design was different, but I think it should be corrected. Also, if the authors want to emphasize the role of GC-MS vs other metabolomics approaches (works that used LC-MS for oral cancer: https://www.nature.com/articles/srep31520, https://www.ncbi.nlm.nih.gov/pmc/articles/PMC5327379/) they should explain the benefits and limitations of GC-MS better.
3) Row 150, “total chromatogram” should be changed to “total ion current”
4) Available “reference compounds” (row 166) should be listed in the Materials section
5) Please use similar style for text organization (margins for a new paragraphs)
6) Since metabolomics is the main part of this work, I think it is necessarily to add more details about the results, such as how many metabolites were detected in total by GC-MS.
7) Row 198 – citric acid is listed twice.

---

## Round 0.2 · Minor Revisions

The reviewers are favorable with the revision and response to comments. As you gather from referee reports, they identified some minor issues. The paper needs to be updated and significantly improved per the recommendations of the reviewers for further editorial consideration.

Reviewer 2 ·

Basic reporting

The authors have made some cosmetic improvements to the MS in their revised version.
Specific comments
Change metabonomics to metabolomics in abstract
Lines 87-92 do not make sense; rephrase.
Add ref for Line 292
Line nos 230 and 308; the conclusions about diagnostic potential is far fetched since this is only a chemical induced animal model. I suggest this be removed.

Experimental design

no comment

Validity of the findings

The lack of independent methods for validation of the results still remains a caveat but may be addressed by other comparative studies.

Reviewer 3 ·

Basic reporting

I would like to thank the authors for their attempts to improve the manuscript according to my and other reviewers suggestions. I think the work has benefited by these changes.

Experimental design

However, I still miss some information:
1) Thank you for clarifying experiment design and providing the exact number of animals per each group. But why PLS-DA image itself does not include all these animals? Do I miss something when see 25 samples (dots) only on the plot in the figure 2?
2) I do not insist on Dunnett’s test (although it is very conventional too), but it is still not clear which two groups were compared by M-W test? Please provide the description and N for each tested group.

Validity of the findings

Conclusions were improved.

Additional comments

Minor points:
end of introduction: appliedto -> applied to
thought the text: space between words and references
if arachidonic acid was confined by external standards, it has to be included to the Method section (so far only citric acid is listed as used external standard)

---

## Round 0.3 · accepted · Accept

The current revision and author’s response to previous comments are acceptable.